# Impact of Oocyte Extract Supplement on Quality of Life after Hepatectomy for Liver Tumours: A Prospective, Multicentre, Double-Blind Randomized Clinical Trial

**DOI:** 10.3390/cancers15102809

**Published:** 2023-05-18

**Authors:** Matteo Donadon, Angela Palmisano, Mariano Bizzarri, Roberto Ceriani, Luigi Veneroni, Gabriele Donati, Davide Tassinari, Massimo Giuseppe Viola, Emiliano Tamburini, Guido Torzilli

**Affiliations:** 1Department of Health Sciences, University of Piemonte Orientale, 28100 Novara, Italy; 2Department of Surgery, University Maggiore Hospital della Carità, 28100 Novara, Italy; 3Department of Hepatobiliary and General Surgery, IRCCS Humanitas Research Hospital, 20089 Rozzano, Italy; 4Systems Biology Group, Department of Experimental Medicine, University La Sapienza, 00161 Rome, Italy; 5Department of Internal Medicine and Hepatology, IRCCS Humanitas Research Hospital, 20089 Rozzano, Italy; 6Department of General and Emergency Surgery, Infermi Hospital, Rimini AUSL Romagna, 47921 Rimini, Italy; 7Department of Internal Medicine, Infermi Hospital, Rimini AUSL Romagna, 47921 Rimini, Italy; 8Department of Oncology, Infermi Hospital, Rimini AUSL Romagna, 47921 Rimini, Italy; 9Department of General Surgery, Cardinale Panico Hospital, 73039 Tricase, Italy; 10Department of Oncology and Palliative Care, Cardinale Panico Hospital, 73039 Tricase, Italy; 11Department of Biomedical Science, Humanitas University, 20090 Pieve Emanuele, Italy

**Keywords:** oocyte extract, quality of life, hepatocellular carcinoma, cholangiocarcinoma, liver surgery, postoperative recovery, supplementation therapy in liver surgery

## Abstract

**Simple Summary:**

Quality of life (QoL) is an important and modifiable concept that should be taken into considerations during treatment allocation in cancer patients. In the last 15 years, studies have reported that supplementation with oocyte extract, which is pivotal during the stage of cell differentiation, may be associated with reduction or suppression of tumour growth and may be, in fact, beneficial in patients with liver tumours. Here, we designed a multicentre, double-blind, randomized clinical trial to assess whether the QoL of patients operated for liver tumours was impacted by receiving a supplement of oocyte extract postoperatively. As shown, the supplement of oocyte extract modifies the QoL after liver surgery by enhancing the functional recovery for many of the QoL items considered. The same result was not recorded with the placebo.

**Abstract:**

Background: Previous studies on oocyte extract supplementation showed benefits in patients with liver tumours. In this trial, we hypothesized that the oocyte extract supplement impacted the QoL after hepatectomy for hepatocellular carcinoma and intrahepatic cholangiocarcinoma. Methods: This was a multicentre, double-blind, randomized clinical trial designed to assess the QoL of patients receiving a supplement of oocyte extract or placebo postoperatively. QoL was assessed using the Short Form-36 questionnaire in participants randomly assigned to treatment (Synchrolevels) or placebo. All study personnel and participants were masked to treatment assignment. The endpoint was the change in the QoL score. Results: Between June 2018 and September 2022, 66 of 128 expected patients were considered as per interim analysis, of which 33 were assigned to the treatment and 33 to the placebo group. Baseline and clinicopathological characteristics were similar between the two groups. In the treatment group, the health, mental and psychological status improved for many of the items considered, reaching statistical significance, while in the placebo group, those items either did not change or were impaired in comparison with the corresponding baseline. Conclusions: Supplementation with oocyte extract modifies QoL after liver surgery by enhancing functional recovery. Further in-depth studies are required to confirm this evidence.

## 1. Introduction

Hepatectomy represents the cornerstone of curative treatment for non-metastatic hepatocellular carcinoma (HCC) and intrahepatic cholangiocarcinoma (iCCA) [1,2,3]. However, surgical resection is associated with physical invasiveness that can be debilitating, especially in elderly patients, and becomes apparent in comparison with other treatments such as percutaneous thermos-ablation, trans-arterial chemoembolization and radiotherapy. Indeed, a proportion of patients may refuse surgery for fear of a longer postoperative recovery period.

Since the 1990s, there has been a burgeoning use of instruments to assess quality of life (QoL) related to different treatments in cancer patients [4,5,6,7,8]. QoL has, in fact, become one of the objective measurements that should be taken into consideration during the decision-making process of treatment allocation in cancer patients [9]. Some previous studies evaluated QoL in patients undergoing hepatectomy, reporting a range from 3 to 12 months to recover to the preoperative baseline level of QoL [10,11,12]. However, these studies were not disease-specific and, being retrospective, were subject to potential selection biases.

The possibility of modulating recovery after hepatectomy is of paramount importance in cancer surgery. In the last 15 years, studies have reported that supplementation with the oocyte extract, which is pivotal during the stage of cell differentiation, may be associated with reduction in or suppression of tumour growth [13,14,15,16]. Oocyte extract may act at the epigenetic level to regulate cell cycle regulators such as p53 and pRb and apoptosis regulators such as E2F-1, c-Myc and p73. More recently, it has been demonstrated that oocyte extract reduces in vitro migration and invasiveness of cancer cells via cytoskeleton remodelling. Furthermore, this extract downregulates the expression of TCTP [17], a protein highly correlated with cancer malignancy: high levels of TCTP promote cell proliferation and cell migration and inhibit apoptosis [18,19]. Further clinical studies reported that oocyte extract supplementation improved the performance status of cancer patients awaiting or undergoing treatment and also provided evidence from a single randomized clinical trial of survival benefits and improvements in performance status outcomes [20,21].

In this study, we hypothesized that the oocyte extract supplement impacted the QoL after hepatectomy for liver tumours. Thus, we planned to conduct a prospective, multicentre, double-blind, randomized clinical trial to assess the QoL of patients receiving a supplement of oocyte extract or placebo postoperatively.

## 2. Materials and Methods

### 2.1. Study Design

This is a prospective, multicentre, randomized clinical trial conducted in three hospitals on a consecutive cohort of patients that underwent hepatectomy for primary liver tumours and received postoperative supplementary therapy with oocyte extract or placebo. The supplemental treatment is produced under the commercial name of Synchrolevels and consists of roe extract, retinol, pyridoxine, Ca pidolate, Mg pidolate and thiamine. The formulation is administered as a sublingual spray. The study protocol (clinicaltrials.gov—registration number NCT05464706) was in accordance with the World Medical Association Declaration of Helsinki and was approved by the Ethical Committee of the leading institution (IRCCS Humanitas Research Hospital, approval number 1910/2018) as well as by the institutional review board of all participating hospitals. Written informed consent was obtained from each patient included in the study. This study followed the Consolidated Standards of Reporting Trials (CONSORT) reporting guidelines [22].

### 2.2. Study Endpoint

The study endpoint was the analysis of the role of supplementary oocyte extract in enhancing recovery after hepatic resection for HCC or iCCA. For this purpose, an established QoL form was adopted at the time of surgery and at 6 months after surgery.

### 2.3. Definitions

The nomenclature and extent of hepatic resection were recorded in accordance with the Brisbane classification [23]. Hepatic resections were considered major when the three adjacent segments were removed. Complications were defined and graded based on the Clavien–Dindo classification [24]. Liver failure was defined and graded based on the definition of the International Study Group of Liver Surgery [25]. Postoperative mortality was recorded 90 days after surgery.

### 2.4. Eligibility Criteria

The inclusion criteria were as follows: presence of written informed consent; age of ≥18 years; histologically proven first diagnosis and first hepatectomy for HCC or iCCA; Eastern Cooperative Oncology Group (ECOG) performance status of 0–1 and normal underlying liver function (Child–Pugh–Turcotte A patients). The exclusion criteria were as follows: refusal to sign the informed consent; age of <18 years; advanced or decompensated underlying cirrhosis; indication to perform adjuvant (postoperative) chemotherapy; postoperative mortality; ECOG > 1; any psychological or psychiatric condition that might compromise the patient’s compliance during the study period.

### 2.5. QoL Assessment

For QoL assessment, we used the Short-Form 36 (SF-36) health status survey version, which is one of the most widely used tools for QoL measurement in cancer patients [26,27,28,29,30]. The SF-36 measures eight different health areas: (1) general health; (2) limitations of activities; (3) physical health problems; (4) emotional health problems; (5) social activities; (6) pain; (7) energy and emotion and (8) general mental health, psychological distress and well-being. The raw data were transformed to norm-based scores for these eight areas to simplify the computation and interpretation of the results, as previously reported [26,27].

### 2.6. Randomization and Masking

Patients were randomly assigned in a 1:1 ratio to either the intervention or the placebo arm. Both patients and clinical investigators were blinded to this assignment. As the random assignment was performed by the supplement manufacturer, the investigators were masked to the randomization sequence.

### 2.7. Surgery

The operations were conducted in the three recruiting centres with similar and consolidated selection criteria, preoperative workup and surgical techniques. Either the laparotomic or the laparoscopic approach was used, and in all patients, the same postoperative protocol—including the enhanced recovery after surgery—was applied. All patients underwent standard follow-up visits every 3 months after surgery by using liver function tests, tumour markers, abdominal ultrasound and/or computed tomography and/or magnetic resonance imaging.

### 2.8. Data Collection

Upon enrolment, the study coordinators of the three centres collected the clinical, histological, surgical and follow-up data using dedicated electronic case report forms, which were then centralized in the leading centre for processing. The QoL forms were collected during the scheduled follow-up visits as hard copies and then centrally digitized for the analysis.

### 2.9. Intervention

The intervention group received the oocyte extract in a sublingual spray formulation at a dose of 1 mL administered three times daily. This treatment was continued under the direct supervision of the study centre coordinator unless discontinued due to poor patient compliance. The control group received a placebo solution in a spray formulation at a dose of 1 mL administered three times daily, which was formulated with the same flavouring and drug packaging as the oocyte extract.

### 2.10. Statistical Analysis

The hypothesis of this trial was that supplementation with oocyte extract would impact the QoL after hepatectomy for liver tumours. By considering 6 months after surgery as the time frame for complete functional recovery after liver surgery [12], we anticipated detection of a mean recovery time of 4.5 months with a standard deviation of 3 months in the intervention group. Therefore, a sample size of 64 patients per arm was requested (α = 0.05; power = 80%). Differences in proportions were analysed using the chi-square test, while differences in distributions were analysed using the t-test or the Mann–Whitney test. A *p* value of less than 0.05 was considered statistically significant. Computations were performed using the IBM-SPSS software Version 22 (SPSS, Chicago, IL, USA). Considering the cost, the resources and the meaningfulness of the trial, an interim analysis was planned after the minimum half of the required patients reached the 6-month follow-up visit [31].

## 3. Results

### 3.1. Patients

Of the 128 expected patients, 66 (51.5%) constituted the first patient cohort to complete the QoL assessment forms before surgery and at 6 months after surgery, which were included in the planned interim analysis. In the same study period, six patients were excluded because they refused to participate or discontinued administration. Table 1 details the baseline demographics and clinical characteristics of these 66 patients. As shown, no significant differences were recorded between the two groups.

### 3.2. Surgical Outcomes

In most cases, a minor hepatectomy performed with the open approach was conducted. In-hospital mortality was nil, and postoperative morbidity was 21.2%. All these cases of postoperative complications were graded as minor morbidity (grade I–II). The median hospital stay after surgery was 7 days (range: 4–12). Table 2 shows the surgical data, which were similar between the two groups apart for an increased use by chance of the laparoscopic approach in the treatment group.

### 3.3. QoL Data

Table 3 details the mean QoL scores for all the SF-36 items in the 33 patients who received the oocyte extract versus those 33 patients who received the placebo, while Figure 1 depicts the changes in such score values. As shown, in the treatment group, not only was the health, mental and psychological status not impaired in any of the considered items, but also, for some items, that status improved. Conversely, in the placebo group, the health, mental and psychological status did not change, or it was impaired in comparison with the corresponding baseline data. As reported in Table 3, for some of the items, the differences between the two timepoints of the analysis reached statistical significance.

## 4. Discussion

This study aimed to clarify the changes in the QoL among patients undergoing hepatectomy for HCC or iCCA treated with or without the oocyte extract supplement. The results showed that the oocyte extract supplement significantly improved postoperative recovery as the mean QoL scores improved after 6 months of treatment in comparison with the respective baseline scores. Such an improvement was not recorded in the placebo group.

Patient QoL has become one of the objective measurements that should be taken into consideration during the decision-making process of treatment allocation in cancer patients [9]. Recently, the European Association of Medical Oncology recommended the use of patient-reported outcomes (PRO) in clinical trials to incorporate the patient’s voice in the evaluation of the risks and benefits of cancer therapies [32]. As a matter of fact, PRO, including QoL reports, are not commonly recorded in clinical trials, especially in the surgical field. In this regard, some changes should be promoted.

Notably, liver surgery may be associated with physical invasiveness that can be debilitating, especially in elderly patients, and becomes apparent in comparison with other treatments such as percutaneous thermoablation, transarterial chemoembolization and radiotherapy, which are emerging as equally effective treatments to surgical resection in selected patients. At the same time, surgical resection still represents the treatment of choice in many HCC or iCCA patients carrying, for instance, large tumours, multiple tumours or tumours with contact or direct infiltration of intrahepatic vascular and biliary structures. Some previous studies evaluated QoL in patients undergoing hepatectomy, reporting a range of 3 to 12 months for recovery to preoperative baseline level of QoL [10,11,12]. Such studies showed that liver surgery may impair the physical QoL temporarily while at the same time improving the mental QoL steadily over time. Of note, when postoperative physical deterioration was reported, it was mainly observed in patients with good preoperative performance status, a relatively young age (i.e., <70 years) and no significant liver tumour burden (i.e., single lesions of <5 cm), indicating that QoL reports should probably be used much more often than they currently are [10,11,12].

Along these lines, the possibility of modulating recovery after hepatectomy by acting on the patient’s QoL areas is of paramount importance in cancer surgery. Here, we tested the role of the oocyte extract following some preliminary intriguing positive experiences [20,21]. The oocyte extract consists of three main protein clusters distinguishable according to their molecular weight: over 45 kDa, around 25–35 kDa and less than 20 kDa. The extract includes multiple forms of yolk protein vitellogenin, heat shock protein, procollagen and polynucleotides such as miRNAs. The oocyte extract has been proven to modulate cell metabolism by inducing a significant shift from a high to a low glycolytic phenotype and a consequent reduction in lactate, fatty acids and citrate synthesis [33]. Furthermore, the oocyte extract induces several mechanisms that promote tumour reversion, i.e., the phenotypic transformation of cancer cells from a malignant to a benign state. Among these mechanisms are p53 activation, re-establishment of normal E-cadherin-based cell-to-cell adhesion and downregulation of translationally controlled tumour protein (TCTP), a protein which has a pivotal role in sustaining cancer proliferation, histone inhibition and chromatin remodelling [34,35]. Although the unknown psychological factors associated to the nutraceutical supportive care cannot be discarded, the administration of fish oocyte extract have been already associated with the improvement of the overall clinical condition. Indeed, oocyte factors contribute to mitigating the impact of chemotherapy-based therapies while increasing their efficacy [36,37], and enhance treatment-induced apoptosis in cancerous tissues [38]. Moreover, as TCTP is usually downregulated during oocyte extract treatments, it can be speculated that this effect would lead to a more efficient utilization of glucose, with concomitant normalization of lipid metabolism [39]. Furthermore, the modulation of TCTP displays stage-specific relevant effects upon liver regeneration [40]. Overall, this evidence suggests that fish oocyte treatment can enact significant metabolic effects.

Certainly, QoL after hepatectomy may be impacted by several different factors. Some authors have already reported positive correlations between liver function and QoL according to the underlying liver disease [41,42,43], while others reported no differences [12]. Similarly, the type and extent of surgery and the postoperative course affect patient QoL, even though it has been reported that good surgical short- and long-term outcomes can counterbalance temporary deterioration of QoL [11,43]. However, in the present study, surgical invasiveness and the postoperative course were well balanced in the two groups.

The limits of the present study may be different to those of previous studies. First, while the number of subjects was statistically calculated and justified a priori, it may be considered too small to allow for a generalization. Second, although the study was an RCT, and patient selection followed the study protocol, the risk of selection bias cannot be completely excluded. Third, we could not investigate the presence of other factors that could have affected the emotional, mental, and social QoL scores of the included patients. Further consideration of these factors remains necessary in the future.

## 5. Conclusions

In conclusion, supplementation with oocyte extract enhances recovery after hepatectomy for HCC or iCCA by acting on several physical and mental items of QoL. Further in-depth studies are required to confirm this evidence.

## Figures and Tables

**Figure 1 cancers-15-02809-f001:**
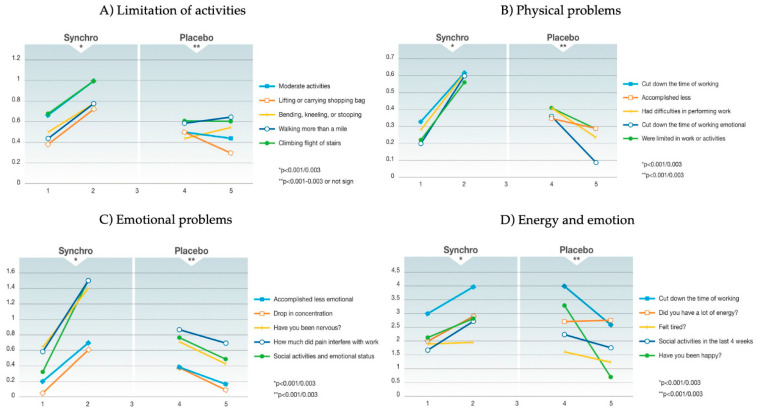
QoL score change. The figure details the graphic representation of the QoL score changes in the treatment group (Synchro) versus the control group (placebo) in the four main domains of the Short-Form 36 health status survey (**A**) limitation in activities; (**B**) physical problems; (**C**) emotional problems; (**D**) energy and emotion). As shown, in the treatment group, the health, mental and psychological status improved for many of the items considered, reaching statistical significance, while in the placebo group, those items either did not change or were impaired.

**Table 1 cancers-15-02809-t001:** Patient characteristics.

Characteristic	Full Series	Treatment Group	Placebo Group	*p*-Value
Patient number	66	33	33	-
Age				
Median; range	70; 27–85	68; 27–74	69; 61–85	0.9144
Sex				
M	51 (77.2)	23 (69.9)	26 (78.7)	
F	15 (22.8)	10 (29.1)	7 (21.3%)	0.7131
Aetiology				
Hepatitis C virus	24 (36.3)	14 (42.4)	10 (30.3)	
Hepatitis B virus	7 (10.6)	3 (9.2)	4 (12.1)	
Alcohol	9 (13.6)	5 (15.1)	4 (12.1)	
Negative	26 (39.5)	11 (33.3)	15 (45.5)	0.6739
Underlying liver				
Chronic hepatitis or cirrhosis	22 (33.3)	9 (27.3)	13 (39.3)	
Normal	44 (66.7)	24 (72.7)	20 (60.7)	0.2962
Pathology				
HCC	55 (83.3%)	26 (78.7)	29 (87.8)	
iCCA	11 (16.7%)	7 (21.2)	4 (12.2)	0.3217
Alpha Fetoprotein				
Median; range	8; 1–82	7; 1–45	8; 3–82	0.9471
Ca19-9				
Median; range	1.2; 0.8–31	2.2; 4–31	2.6; 0.8–22	0.8719
Platelet count				
Median; range	187; 97–282	154; 97–127	131; 111–282	0.7194
CPT score A	66 (100)	33 (100)	33 (100)	-
MELD				
Median; range	7; 6–17	6; 6–11	7; 6–17	0.9181
Tumour size (cm)				
Median; range	4; 1–11	4; 1–7	3.5; 1–11	0.8740
Tumour number				
Median; range	1; 1–3	1; 1–3	1; 1–2	0.8981
Vascular Invasion				
Micro	35 (53)	25 (75.7)	8 (24.2)	
Macro	9 (13.6)	4 (12.1)	5 (15.1)	0.0711
Grading				
1–2	28 (42.4)	11 (45.4)	21 (63.6)	
3–4	35 (53.1)	19 (72.7)	12 (36.3)	
Unknown	3 (4.5)	3 (9)	-	**0.0321**

Notes and abbreviations: HCC, Hepatocellular carcinoma; iCCA, Intrahepatic cholangiocarcinoma; CPT, Child–Pugh–Turcotte; MELD, Model for End-stage Liver Disease. Bold data indicates statistically significant values.

**Table 2 cancers-15-02809-t002:** Surgical data.

Characteristic	Full Series	Treatment Group	Placebo Group	*p*-Value
Extent of hepatectomy				
Major (>3 segments)	11 (16.6)	5 (15.1)	6 (18.1)	
Minor	55 (83.4)	28 (84.9)	27 (81.9)	0.7411
Approach				
Open surgery	56 (84.8)	25 (75.7)	31 (94)	
Laparoscopic surgery	10 (15.2)	8 (24.3)	2 (6)	**0.0394**
Length of operations (minutes)				
Median; range	314; 96–654	213; 96–234	296; 108–654	0.9341
Length of Pringle maneuver				
Median; range	27; 0–145	22; 0–81	18; 0–145	0.6714
Blood loss (mL)				
Median; range	200; 0–1400	180; 0–340	250; 0–1400	0.8713
Red packed cell transfusion	11 (16.6)	5 (15.1)	6 (18.1)	-
Postoperative complications				
Overall	14 (21.2)	9 (27.2)	6 (18.1)	
Clavien–Dindo 1–2	14 (21.2)	-	-	
Clavien–Dindo 3–4	-	-	-	-
Length of stay (day)				
Median; range	7; 4–12	7; 5–12	6; 4–9	0.5618
90-day mortality	-	-	-	-

Notes: Bold data indicate statistically significant values.

**Table 3 cancers-15-02809-t003:** Summary of SF-36 QoL reports.

Item	Treatment Group (N = 33)			Placebo Group (N = 33)		
	Baseline	At 6 Months	*p*-Value	Baseline	At 6 Months	*p*-Value
General health	2.76 ± 0.56	2.94 ± 0.66	0.1530	3.33 ± 1.02	2.89 ± 0.91	**0.0281**
General health, compared to one year ago	3.00 ± 0.94	3.94 ± 1.09	**<0.001**	4.06 ± 0.98	2.61 ± 1.11	**<0.001**
Limitation of activities						
Vigorous activities	1.22 ± 0.62	1.44 ± 0.65	0.0930	0.94 ± 0.77	0.75 ± 0.86	0.2570
Moderate activities	0.67 ± 0.77	1.00 ± 0.77	**0.0384**	0.5 ± 0.79	0.44 ± 0.78	0.7089
Lifting or carrying shopping bag	0.39 ± 0.57	0.72 ± 0.61	**0.0074**	0.5 ± 0.71	0.39 ± 0.78	0.4718
Climbing flights of stairs	0.68 ± 0.49	1.00 ± 0.69	**0.0082**	0.61 ± 0.78	0.61 ± 0.85	1
Climbing one flight of stairs	0.22 ± 0.59	0.33 ± 0.43	0.2992	0.33 ± 0.59	0.33 ± 0.69	1
Bending, kneeling, or stooping	0.5 ± 0.73	0.78 ± 0.62	**0.0457**	0.44 ± 0.7	0.5 ± 0.79	0.6946
Walking more than a mile	0.44 ± 0.81	0.78 ± 0.62	**0.0231**	0.59 ± 0.71	0.65 ± 0.86	0.7102
Walking several blocks	0.35 ± 0.61	0.35 ± 0.49	1	0.5 ± 0.62	0.33 ± 0.69	0.2073
Walking one block	0.18 ± 0.39	0.18 ± 0.39	1	0.28 ± 0.57	0.17 ± 0.51	0.3216
Bathing or dressing yourself	0.06 ± 0.24	0.00 ±0.00	1	0.17 ± 0.38	0.11 ± 0.47	0.4933
Physical problems						
Cut down on the time of working	0.33 ± 0.5	0.61 ± 0.49	**0.0064**	0.41 ± 0.51	0.24 ± 0.44	0.0836
Less accomplishment	0.22 ± 0.51	0.56 ± 0.43	**0.0006**	0.35 ± 0.49	0.29 ± 0.47	0.5419
Limitation in work or activities	0.22 ± 0.51	0.56 ± 0.43	**0.0006**	0.41 ± 0.51	0.29 ± 0.47	0.2336
Difficulties in performing work	0.28 ± 0.5	0.61 ± 0.46	**0.0011**	0.41 ± 0.51	0.24 ± 0.44	0.2336
Emotional problems						
Cut down on the time of working	0.20 ± 0.52	0.60 ± 0.42	**<0.001**	0.36 ± 0.5	0.09 ± 0.3	**0.0018**
Less accomplishment	0.22 ± 0.46	0.71 ± 0.43	**<0.001**	0.41 ± 0.51	0.18 ± 0.39	**0.0148**
Drop in concentration	0.06 ± 0.5	0.61 ± 0.24	**<0.001**	0.39 ± 0.5	0.11 ± 0.32	**0.0015**
Social activities and emotional status	0.33 ± 0.79	1.51 ± 0.69	**<0.001**	0.78 ± 1.06	0.5 ± 0.92	0.1702
Pain in the last 4 weeks	1.41 ± 1.33	1.12 ± 1.27	0.2774	0.72 ± 1.02	0.44 ± 1.04	0.1862
How much pain interfered with work	0.65 ± 0.87	1.41 ± 0.93	**<0.001**	0.56 ± 0.86	0.39 ± 0.98	0.3687
Energy and emotion in the last 4 weeks						
Full of pep?	2.38 ± 1.31	2.75 ± 1	0.1232	3.17 ± 1.49	2.28 ± 1.38	**0.0031**
Have you been nervous?	0.59 ± 1.1	1.5 ± 0.48	**<0.001**	0.88 ± 0.93	0.71 ± 1.26	0.4539
Felt down?	0.88 ± 1.02	0.69 ± 1.01	0.3615	0.59 ± 0.87	0.53 ± 0.87	0.7362
Felt calm and peaceful?	2.69 ± 1.01	3 ± 1.26	0.1742	3.47 ± 1.5	3.24 ± 1.71	0.4853
Did you have a lot of energy?	2 ± 1.26	2.88 ± 1.02	**0.0003**	2.71 ± 1.71	2.65 ± 1.46	0.8542
Felt downhearted?	1.24 ± 0.75	1.06 ± 1.25	0.3945	0.94 ± 0.9	0.59 ± 0.8	0.0469
Did you feel worn out?	0.81 ± 1.05	0.63 ± 1.09	0.4120	0.59 ± 0.71	0.76 ± 0.9	0.3069
Have you been happy?	2.13 ± 1.19	2.8 ± 1.21	**0.0041**	3.29 ± 1.31	0.71 ± 1.16	**0.0239**
Felt tired?	1.88 ± 0.78	1.94 ± 0.97	**<0.001**	1.65 ± 1.27	1.24 ± 1.2	0.1074
Social activities in the last 4 weeks	1.65 ± 1.11	2.71 ± 0.69	**<0.001**	2.24 ± 0.9	1.76 ± 1.03	**0.0169**
General health						
I seem to get sick easier than others	2.24 ± 1.37	2.54 ± 1.2	0.4359	2.25 ± 1.39	1.75 ± 0.93	**0.0411**
I am as healthy as anybody	2.94 ± 1.34	3 ± 1.46	0.8343	3 ± 1.1	2.94 ± 1.44	0.8191
I expect my health to get worse	2.59 ± 1.33	2.47 ± 1.28	0.6535	2.81 ± 1.42	2.19 ± 0.98	**0.0145**
My health is excellent	2.81 ± 1.38	3.19 ± 1.33	0.1728	3.56 ± 1.31	2.88 ± 1.09	**0.0069**

Notes: Bold values indicate statistically significant data.

## Data Availability

The analysis was performed by using data from adult patients enrolled in the study and was conducted according to the guidelines of the Declaration of Helsinki. The data sets generated and/or analysed during the current study are not publicly available but are available from the corresponding author on reasonable request.

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
