# Peer review of "Impact of Oocyte Extract Supplement on Quality of Life after Hepatectomy for Liver Tumours: A Prospective, Multicentre, Double-Blind Randomized Clinical Trial"

_cancers, 2023, doi:10.3390/cancers15102809_

Round 1

Reviewer 1 Report

The authors presented in the present paper an interesting research on the impact of oocyte extract on QoL in patients after liver surgery. Despite the relatve fast anatomic recovery the return to full QoL as before surgery remains an obstacle. The results sugested that this can be improved with oocyte extract. the paper is well written and easy to read.

There are some remaining questions:

First: to me it is unclear what a p-value >than 1 in table 1 means. As I understand p is between 0 and 1. Can you explain this.

Second: In the discussion you talked about the effect of oocyte extract on tumor biology. I think in this study the effect on regeneration is probably more of concern. What are the mechanism here? Further, are there more general psychologic or physiologic effects known witch influence well being in gerneral?

Third: How many patients suffered from a recurrence of the tumor in each group? Did this have an impact on QoL?

Author Response

R1:

The authors presented in the present paper interesting research on the impact of oocyte extract on QoL in patients after liver surgery. Despite the relative fast anatomic recovery the return to full QoL as before surgery remains an obstacle. The results suggested that this can be improved with oocyte extract. the paper is well written and easy to read.

Reply: We thank the reviewer for her/his time in doing this work.

There are some remaining questions:

First: to me it is unclear what a p-value >than 1 in table 1 means. As I understand p is between 0 and 1. Can you explain this.

Reply: We thank for this observation, and we are sorry for those typos. Those values more than 1 represented the values of the test, for instance the Chi-square values, rather than the p-values. We have now corrected the values in Table 1. Again, sorry.

Second: In the discussion you talked about the effect of oocyte extract on tumor biology. I think in this study the effect on regeneration is probably more of concern. What are the mechanisms here? Further, are there more general psychologic or physiologic effects known which influence well-being in general?

Reply: We thank for this note, which is important. We have added the following paragraph in the discussion:

Albeit unknown psychological factors associated to the nutraceutical supportive care cannot be discarded, the administration of           fish oocyte  extract have been already associated with the improvement of the overall clinical condition. Indeed,      oocyte factors contribute in mitigating the impact of chemotherapy-based therapies, while increasing their efficacy ([1]-2), and enhance treatment-induced apoptosis in cancerous tissues (3). Moreover, as TCTP is usually downregulated during       oocyte extract treatments, it can be speculated that this effect would lead to a more efficient utilization of glucose, with concomitant normalization of lipid metabolism (4). Furthermore, modulation of TCTP display stage-specific relevant effects upon liver regeneration (5). Overall, this evidence suggests that      fish oocyte extract treatment can enact significant metabolic effects.

References:

  1. Proietti S, Cucina A, Catizone A, Ricci G, Pensotti A, Bizzarri M. Zebrafish embryo extracts enhance 5-FU anti-cancer effects upon breast cancer cells. Eur Rev Med Pharmacol Sci. 2021 Apr;25(8):3235-3245. doi: 10.26355/eurrev_202104_25732. PMID: 33928609.
  2. Proietti, S,Cucina, A, Giuliani, A, Verna, R, Palombi, E, Biava, PM, Pensotti, A, 2018, “Fish protein extract enhances clinical response to salvage chemotherapy in colon cancer patients“ Organisms. Journal of Biological Sciences, vol. 2, no. 2, pp. 81-90. DOI: 10.13133/2532-5876_4.8
  3. Anselmi F, Cucina A, Biava PM, Proietti S, Coluccia P, Frati L, Bizzarri M. Zebrafish stem cell differentiation stage factors suppress Bcl-xL release and enhance 5-Fu-mediated apoptosis in colon cancer cells. Curr Pharm Biotechnol. 2011 Feb 1;12(2):261-7. doi: 10.2174/138920111794295864. PMID: 21043999.
  4. Dong K, Zhao Q, Xue Y, Wei Y, Zhang Y, Yang Y. TCTP participates in hepatic metabolism by regulating gene expression involved in insulin resistance. 2021 Feb 5;768:145263. doi: 10.1016/j.gene.2020.145263. Epub 2020Oct 26. PMID: 33122078.
  5. Zhu WL, Cheng HX, Han N, Liu DL, Zhu WX, Fan BL, Duan FL. Messenger RNA expression of translationally controlled tumor protein (TCTP) in liver regeneration and cancer. Anticancer Res. 2008 May-Jun;28(3A):1575-80. PMID:18630514.

The new text with the dedicated references has been added in the discussion section.

Third: How many patients suffered from a recurrence of the tumor in each group? Did this have an impact on QoL?

Reply: We thank for this question. As reported in the manuscript, this paper is an interim analysis done on half of the patients theoretically requested for the study. And here we focused only on the primary endpoint of the study, which was the changes in QoL. We also set a secondary endpoint on the oncological outcome, which will be analyzed once all the enrolled patients will have completed the 2-year follow-up. While the disease recurrence is very important, this study was statistically thought and designed for the QoL endpoint, which means that either a negative finding or a positive finding regarding a change in the disease recurrence risk will require a cautious interpretation. However, we are also very curious about the oncological outcome.

Reviewer 2 Report

In present manuscript, Donadon et al. designed a multicentre, double-blind, randomized clinical trial to assess the impacts of oocyte extract on the quality of life (QoL) in patients operated for liver tumors. The author presented data showed that the supplement of oocyte extract modifies the QoL after liver surgery by enhancing the health, mental and psychological status while placebo group showed no changes or were impaired in comparison with the corresponding baselines. The author thus concluded that supplementation with oocyte extract modifies QoL after liver surgery by enhancing functional recovery. Overall the author presented insteresting findings might benefit clinical practice. However, the following concerns need addressed.

1.    English needs to be carefully revised and improved in spelling and style.

2.    Caption of table and figure need be more specified.

3.    When compare the surgical approach of treatment and placebo groups, 2 patients in placebo group, while 8 patients in treatment group were received laparoscopic surgery. Whether the differential treatment led to the differences in recovery need to be addressed.

4.    Several indicators in table 3 including have you been happy, emotional problems showed opposite trends, how to interpret these results?

5.    Table 3, the author only compared statistical differences before and after treatment, differences between groups were also preferred.

6.    Did the author performed any other medical measurement besides QoL forms to evaluate patients’ recovery?

Author Response

In present manuscript, Donadon et al. designed a multicentre, double-blind, randomized clinical trial to assess the impacts of oocyte extract on the quality of life (QoL) in patients operated for liver tumors. The author presented data showed that the supplement of oocyte extract modifies the QoL after liver surgery by enhancing the health, mental and psychological status while placebo group showed no changes or were impaired in comparison with the corresponding baselines. The author thus concluded that supplementation with oocyte extract modifies QoL after liver surgery by enhancing functional recovery. Overall the author presented insteresting findings might benefit clinical practice. However, the following concerns need addressed.

Reply: We thank the reviewer for her/his time in doing this work.

  1. English needs to be carefully revised and improved in spelling and style.

Reply: Thank you for this note. We went through the manuscript to improve the English. However, please be advised that the manuscript has been proofread by and independent agency that has also provided a certificate of editing available as a supplementary document.

  1. Caption of table and figure need be more specified.

Reply: We thank for this observation, and we are sorry for those typos that have been now fixed.

  1. When compare the surgical approach of treatment and placebo groups, 2 patients in placebo group, while 8 patients in treatment group were received laparoscopic surgery. Whether the differential treatment led to the differences in recovery need to be addressed.

Reply: We thank for this observation. It is true that more patients in the treatment group had laparoscopic surgery. Such difference was by chance, and it was not planned. This is the only difference between the two groups in terms of surgical data as depicted by table 2. Consistently with this right observation, we have now added a sentence in the results section. However, please be advised that the same postoperative protocol – including the standard enhanced recovery after surgery (ERAS) – was applied in all the patients independently by the use of the laparoscopy. This information has been now added in the manuscript.

  1. Several indicators in table 3 including have you been happy, emotional problems showed opposite trends, how to interpret these results?

Reply: We thank for the opportunity to clarify this aspect. While for sure it is difficult to estimate the psychological components of the patient health, which depends to many different issues not only related to the surgical procedure, in most of the cases and in most of the items of the SF36 QoL form the global tendency was in favor of the treatment indicating an overall enhanced functional recovery in that group of patients receiving Synchrolevels. Certainly, more deeper studies should be planned to confirm our preliminary evidence. 

  1. Table 3, the author only compared statistical differences before and after treatment, differences between groups were also preferred.

Reply: We thank for this note. Being a randomized clinical trial important difference between the two groups were not present as depicted in Table 1 and 2. As rightly said before, the only difference in surgical data was about the rate of the laparoscopic approach, which was increased in the treatment group. Thus, we focused the analyses on difference in QoL before and after the operation associated to Synchrolevels administration.

  1. Did the author performed any other medical measurement besides QoL forms to evaluate patients’ recovery?

Reply: In this study we used the SF36 QoL form and we statistically designed all the study those items. The sample size calculation and all the subsequent analyses were planned on that form. And this was done because the primary endpoint of the study was the changes in QoL. Other types of patient-reporting outcomes were not used. Additionally, we had the data on postoperative courses as collected during the postoperative in-hospital period and during the follow-up. Of note, no differences in these data were recorded indicating a similar postoperative course between the two groups.

Round 2

Reviewer 2 Report

The author have addressed all concerns of this reviewer.